# DHA-FL: Enabling Efficient and Effective AIoT via Decentralized Hierarchical Asynchronous Federated Learning

**Houston Huff** [*1]   **Pinyarash Pinyoanuntapong** [*1]   **Ravikumar Balakrishnan** [2]   **Hao Feng** [2]   **Minwoo Lee** [1]
**Pu Wang** [1]   **Chen Chen** [3]

## ABSTRACT

The challenges of scalability, robustness, and resilience to slow devices have posed significant obstacles to the effective and efficient implementation of Federated Learning (FL), a crucial technology for the emerging Artificial Intelligence of Things (AIoT). This paper proposes a solution to these challenges with the introduction of a Decentralized Hierarchical Asynchronous Federated Learning Scheme (DHA-FL). This scheme utilizes a hierarchical edge computing architecture, enabling a two-stage model aggregation paradigm that significantly enhances system scalability. To further enhance system robustness, decentralized asynchronous model aggregation is adopted among edge servers to prevent single node failures while mitigating the impact of slow devices or stragglers. Our experiments, conducted on a live wireless multi-hop IoT testbed, demonstrate that DHA-FL can achieve convergence in approximately half the time compared to the centralized hierarchical approach. Moreover, it enables an even more significant convergence speed-up (up to 8x) over the classic FedAvg baseline when dealing with stragglers.

## 1 INTRODUCTION

In recent years, the applications of artificial intelligence (AI) have greatly expanded. The Internet of Things (IoT) (Atzori et al., 2010), which comprises networks of connected, cooperative devices, offers a fertile ground for leveraging AI to analyze local data and make decisions based on independent analysis and peer communication. This is referred to as the Artificial Intelligence of Things (AIoT) and has a wide range of applications, such as traffic control, security, healthcare, and living assistance, to name a few (Zhang & Tao, 2020). However, these applications and others require a vast and ever-increasing volume of data and training devices for delivering accurate machine learning based solutions. As the volume of data increases, the strain on the network due to traditional centralized learning solutions increases tremendously, necessitating new networking and learning paradigms.

Federated Learning (FL) (Konečný et al., 2016) has proven itself to be greatly communication-efficient over central-ized learning. Rather than extracting local data, each edge server takes on the responsibility of training its own local data and sharing its own machine learning model. FL fully utilizes the computational resources of the edge servers and improves security by avoiding direct sharing of data. FL, typically, can take the form of either centralized federated learning (CFL), where the IoT workers send their trained models to the central server for aggregation to a global model, or decentralized federated learning (DFL) (Lian et al., 2017; Cao et al., 2021), where each IoT worker first updates its local model and then averages its local model only with its immediate neighbors.

Although Federated Learning (FL) has proven to be effective, it still faces a number of challenges. CFL is hindered by robustness concerns due to a single point of failure, which DFL alleviates. However, scalability remains a limiting factor for both CFL and DFL due to the inherent communication bottlenecks experienced as the number of IoT workers increases. Specifically, CFL concentrates all its communication traffic load around a central server, while DFL's two-way peer-to-peer communication among IoT devices becomes expensive, particularly in interference-rich wireless environments. Furthermore, both CFL and DFL typically assume ideal hardware and network performance, without accounting for slow workers or stragglers, which can significantly impede model convergence speed.

In this paper, we introduce a Decentralized Hierarchical Asynchronous Federated Learning (DHA-FL) approach,

---

[*]Equal contribution [1]College of Computing and Informatics, University of North Carolina at Charlotte, Charlotte, North Carolina, United States [2] Intel Labs, United States [3]University of Central Florida, Orlando, Florida, United States. Correspondence to: Houston Huff <whuff1@uncc.edu>, Pinyarash Pinyoanuntapong <ppinyoan@uncc.edu>.

*Proceedings of the 6th MLSys Conference Workshop on Resource-Constrained Learning in Wireless Networks*, Miami, FL, USA, 2023. Copyright 2023 by the author(s).

designed to simultaneously address robustness, scalability, and straggler challenges in the federated learning process. By incorporating a decentralized hierarchical network structure, DHA-FL employs a two-stage model aggregation paradigm to enhance system scalability and robustness. First, each edge server maintains and updates its edge model by aggregating local models from its assigned IoT workers using a centralized FL paradigm. Second, the edge servers then update the shared global model through decentralized model aggregation in which each server averages the edge models received from neighboring edge servers. To combat the impact of stragglers, the edge servers adopt asynchronous model averaging, allowing each edge server to opportunistically aggregate currently received edge models from its neighbors without waiting for slow edge servers affected by stragglers or disruption by neighbor failures.

Not only does DHA-FL offer a powerful solution to stragglers, scalability, and robustness, but also it does so with a generic, lightweight framework that incurs minimal overhead, making it a great tool also for development purposes. Furthermore, we have tested this framework in a live multi-hop IoT network, providing a more accurate representation of real-world scenarios for FL experiments than simulated environments. The majority of existing research conducts FL experiments in simulation or ones evaluated assuming one-hop communication on Wi-Fi access points or cellular base stations, which may not accurately reflect the networking conditions of real multi-hop IoT networks.

## 2 RELATED WORK

DFL paradigms, known for their resilience to single-node failures, predominantly utilize synchronous optimization strategies (Lian et al., 2017; Cao et al., 2021) (Sync-DFL). The Collaborative FL algorithm (Chen et al., 2020) employs a hybrid technique that combines CFL and Sync-DFL. In this method, workers initially engage in learning through a central server, but if the server becomes inaccessible, they transition to a one-hop neighbor Sync-DFL. Although Sync-DFL offers improved robustness, it is vulnerable to stragglers (i.e., slow workers), leading to a considerable reduction in overall model convergence speed. To address this issue, Async-DFL has been proposed in (Pinyoanuntapong et al., 2022; Lian et al., 2018). However, this method assumes a flat network topology, preventing it from utilizing edge computing nodes (e.g., Wi-Fi routers and cellular base stations) for scalable FL. Hierarchical FL (HFL) (Abad et al., 2020) adopts a centralized hierarchical FL paradigm, exhibiting improved communication efficiency and system scalability compared to traditional CFL solutions. Nevertheless, HFL still relies on a central server to conduct global model updates and operates in a synchro-

nized model aggregation mode and consequently suffers all of CFL's core shortcomings. Most importantly, none of the aforementioned solutions have been implemented and tested on real-life wireless systems.

## 3 DHA-FL

The DHA-FL solution summarized in Algorithm 1 and 2 is illustrated in Fig. 1. The learning network is composed of a number of edge servers, each of which serves multiple IoT workers. The edge servers are interconnected to each other via a wireless mesh network.

Each edge server begins its training task with identical untrained models. The edge servers first propagate their models to their respective $K$ worker nodes (Step 1 of Fig. 1), which then perform multiple local rounds of batched SGD training (Step 2 of Fig. 1), defined as $w_k = w_k - \eta \frac{1}{\beta} \sum_{x_k \in X_k} \Delta f(w_k; x_k)$, where $X_k$ is a mini-batch of the training samples of worker $k$, and $\beta$ is the batch size. Once training is complete, the workers send their trained models back to their corresponding edge server. Once all the worker models are received, the edge server performs the weighted average on the received models to yield the updated edge model (Step 3 of Fig. 1), expressed as:

$$w^{(E)} = \sum_{k=1}^{K} \lambda_k w_k \tag{1}$$

where $w^{(E)}$ is the updated edge server model, $K$ is the number of workers covered by the edge server, $w_k$ is the model weights of worker $k$, and $\sum_{k \leq K} \lambda_k = 1$.

Once an edge server completes the aforementioned local model aggregation process, it broadcasts its edge model $w^{(E)}$ to its neighboring edge servers (Step 4 of Fig. 1). Then, it performs edge model aggregation by averaging the models in its model queue, i.e.,

$$w^{(E)} = \sum_{n=1}^{|W^{(E)}|} \frac{w_n^{(E)}}{|W^{(E)}|} \tag{2}$$

and then the aggregated edge model is sent back to its served IoT workers (Step 5 of Fig. 1).

Each edge server maintains its own model queue, denoted as $W^{(E)}$, which includes its own edge model and those it receives opportunistically from neighboring servers. The edge model aggregation process operates asynchronously, meaning the server doesn't have to wait for all neighboring servers to send their models. As a result, faster servers can continue without waiting for slower ones. Since this process is asynchronous, each server tracks its own training epoch count up to a predetermined number. The five

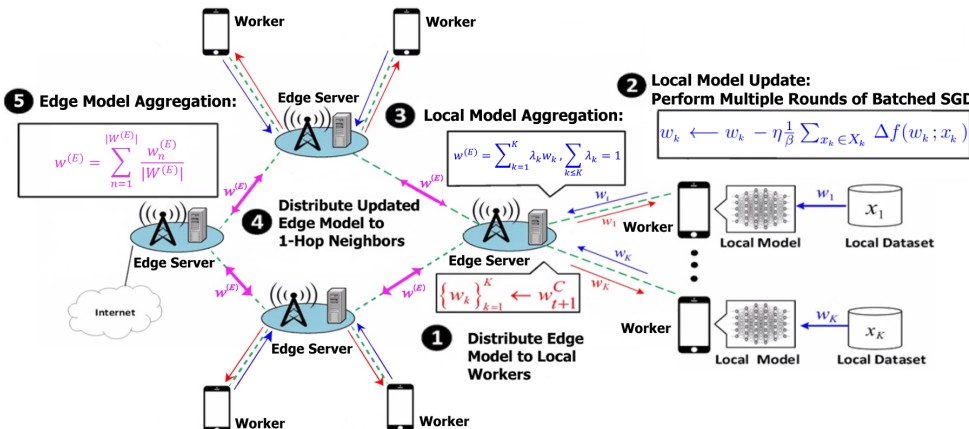

*Figure 1.* DHA-FL utilizes the edge servers as intermediate aggregators for the worker devices. The edge servers distribute their models to their workers, then after the workers finish training, each edge server collects and averages the worker models into the updated edge model. The edge servers then broadcast their respective updated models to their 1-hop neighbors and the ones they receive are aggregated again with their own updated model locally.

steps outlined in the previous section constitute one training epoch for each edge server.

For final consensus in network models, a final-round global model aggregation is performed. This process involves each edge server sending its edge model to the global aggregation server, which performs model averaging to produce the final inference model used by IoT devices.

---

**Algorithm 1** DHA-FL Edge Server

1: **function** LocalModelAggregation
2:    **for** each worker $k$ in $K$ **do**
3:       Distribute $w_k$ to $k$
4:       UpdateWorkerModel($k$)
5:    **end for**
6:    $w^{(E)} \longleftarrow \sum_{k=1}^{K} \lambda_k w_k, \sum_{k \leq K} \lambda_k = 1$
7: **end function**
8: **function** EdgeModelAggregation
9:    $W^{(E)} \longleftarrow$ append $w^{(E)}$
10:   broadcast model $w^{(E)}$ to neigbor edge servers
11:   **if** model $w_n^{(E)}$ arrives **then**
12:      $W^{(E)} \longleftarrow$ append $w_n^{(E)}$
13:   **end if**
14:   $w^{(E)} \longleftarrow \sum_{n=1}^{|W^{(E)}|} \frac{w_n^{(E)}}{|W^{(E)}|}$
15:   $W^{(E)} \longleftarrow \emptyset$           // Empty Buffer
16: **end function**

---

## 4    EXPERIMENT SETUP

In this section, we present the real-world IoT network testbed in details. Then, we describe the models, datasets, and baselines used in the experiments.

**Physical Testbed Setup**

---

**Algorithm 2** DHA-FL Worker

1: **function** UpdateWorkerModel($k$)
2:    **for** local training epoch $h = 1, 2, ..., H$ **do**
3:       **for** each local batch b $\in \beta$ **do**
4:          $w_k \longleftarrow w_k - \eta \frac{1}{\beta} \sum_{x_k \in X_k} \Delta f(w_k; x_k)$
5:       **end for**
6:    **end for**
7:    Send $w_k$ to edge server
8: **end function**

---

For our experiments, we have opted to employ an edge computing network system, consisting of 27 virtual workers for local training, 10 edge servers for local aggregation, and 10 wireless-mesh routers for backbone communication. We have leveraged an edge computing network to model an IoT network, as illustrated in Figure 2. Our testbed includes 10 wireless edge servers, each with a wireless embedded router for communication and an Nvidia Xavier node for computation. The Nvidia Xavier node is equipped with Ubuntu 20.04 operating system and 16GB combined RAM for GPU and CPU. To enable each Jetson edge server to support a different number of virtual FL workers (in our case, 3), we have utilized network namespace to isolate TCP/IP layer within each edge server. To facilitate multi-radio wireless communication, we have equipped each wireless router with three wireless interface cards, which operate in MeshPoint (MP) mode with a fixed 2.4 GHz and 5 GHz channel and 20 MHz channel width in 802.11ac operating mode, with a transmit power of 15 dBm. We utilized the state-of-the-art Batman-adv protocol (Marek Lindner, 2011) for multi-hop routing. This establishes the server-to-worker connection in the FL setting.

**Heterogeneous Data Settings and Models**

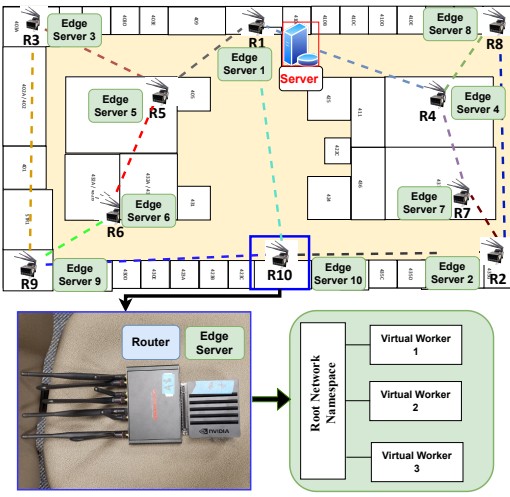

*Figure 2.* IoT Multi-hop Network topology.

Our experiments were conducted with the datasets FEMNIST with 62 classes from LEAF (Caldas et al., 2018; lea, 2019) and CIFAR-10 (Krizhevsky, 2009; cif) with 10 classes. We followed the non-IID data settings with realistic partition method from LEAF. For CIFAR-10, we use the Dirichlet distribution $Dir(\beta)$ to create heterogeneous data partitions for all the workers in unbalanced settings. The degree of heterogeneity was chosen to be the value ($\beta$ = 0.5). Our convolutional neural network (CNN) is composed of two convolutional layers followed by a fully connected layer that utilize local SGD. The convolutional layers contain 32 and 64 layers respectively and are attached via a 2x2 max pooling layer. The fully connected layer contains 128 units with ReLU activation and outputs into the final layer as fully connected with softmax activation, whose file size is around 10 MBytes, represents a high communication volume. We also evaluated the CIFAR-10 dataset using a lightweight version of the deep neural network model, MobileNet (Howard et al., 2017), with a width multiplier of ($\alpha$ = 0.25). This reduction in the size of the network models was necessary to support multiple virtual workers deployment on a single Jetson device, which has limited GPU resources. The size of the resulting model is about 2 MBytes, which represents a lower communication volume.

## FL Implementations

In order to conduct our experiments, we utilized the DHA-FL experiential framework to implement the classic flat centralized FL (CFL), hierarchical centralized FL (HCFL), hierarchical synchronous decentralized FL (DHS-FL), and hierarchical asynchronous decentralized FL (DHA-FL). The experiments involving CFL utilized the widely-adopted FedAvg (He et al., 2020) algorithm. Each experimental run involved training with 5 local rounds, at least 20 global rounds, a batch size of 10, and a learning

rate of 0.001 for CIFAR-10 and 0.02 for FEMNIST. To simulate computational and communication heterogeneity, we introduced straggler workers by adding a delay of 40 seconds per local epoch to the first worker of specified edge servers, enabling us to study the impact of straggler workers on the performance and scalability of the different hierarchical solutions and providing valuable insights into the behavior of these systems in real-world scenarios.

## Performance Comparisons

We evaluate the model convergence by observing the learning curves and the wall-clock time when the testing accuracy achieves certain thresholds (0.4, 0.5, and 0.6) where all methods can achieve the same maximum accuracy of roughly 0.65 as shown in Figure 3.

In the absence of stragglers in Figure 3(a), all of the FL-based hierarchical solutions were found to be particularly effective, with both HCFL, DHS-FL, and DHA-FL outperforming the flat CFL baseline by approximately 1.5x and 2.4x, respectively faster in term of training time as shown in Figure 3(d). This is because hierarchical FL-based solutions are able to reduce the amount of costly FL communication traffic by aggregating local models at the edge server before sending them to the neighbors or the central server. When a single straggler is introduced as shown in Figure 3(b), all modes directly experience increased training time; CFL, HCFL, and DHS-FL all suffer most as one slow edge server keeps the remaining edge servers waiting and reach the same accuracy of 0.6 by about 100, 70, and 59 minutes, respectively, whereas DHA-FL is significantly less affected as the remaining edge servers continue their own training opportunistically and converge to the same point of accuracy of 0.60 in 22 minutes, 4x faster than the CFL baseline.

## Convergence Time over various numbers of stragglers and locations

We conducted a study on the impact of the number and location of stragglers as shown in Table 1, summarizing the convergence time for all methods to achieve the maximum testing accuracy. The percentage of stragglers is a rounded percentage of the edge servers that are affected by slow workers (stragglers). The runtime difference between CIFAR-10's MobileNet model and FEMNIST's CNN model is attributed to the differences in their computational training costs and communication cost, e.g., the model file size is about 2MB for MobileNet and about 10MB for CNN. The overall trends seen across our FL methods remain consistent across both cases. With increasing straggler density, CFL, HCFL, and DHS-FL rapidly approach a bound of performance loss for the given straggler time delay, as shown in Table 1. In contrast, for DHA-FL the convergence speed worsens at a reduced, more proportional rate to the straggler density. The most egregious dis-

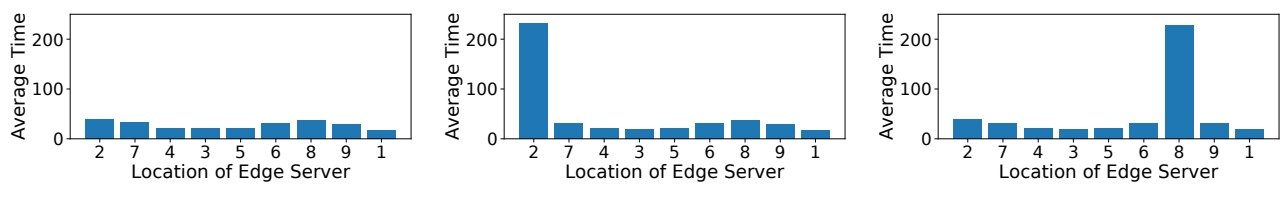

(a) Avg Local Training Time (seconds) with No Straggler

(b) Avg Local Training Time (seconds) with straggler at Edge-server 2

(c) Avg Local Training Time (seconds) with straggler at Edge-server 9

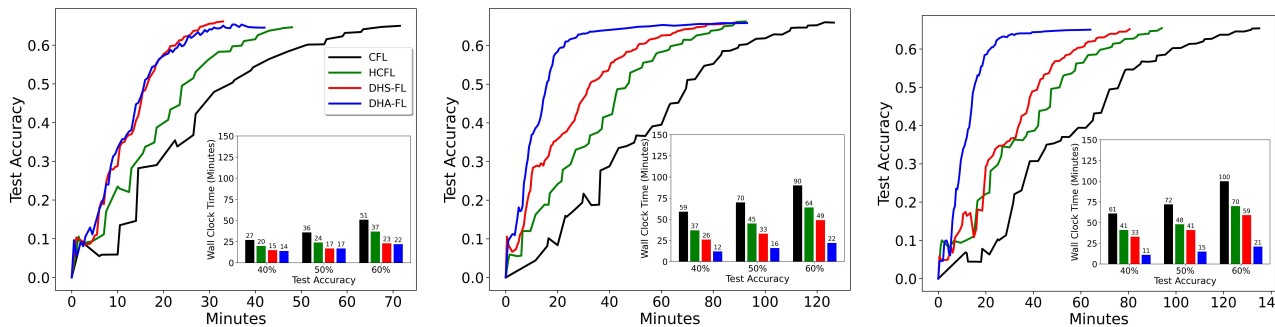

(d) FEMNIST Wall Time Convergence Accuracy with No Straggler

(e) FEMNIST Wall Time Convergence Accuracy with straggler at Edge-server 2

(f) FEMNIST Wall Time Convergence Accuracy with with straggler at Edge-server 9

*Figure 3.* Comparison of CFL, HCFL, DHS-FL, and DHA-FL learning performance and average training time in testbed environment (CIFAR-10 and FEMNIST). The comparisons show cases for no straggler, a straggler on the R3 device, and straggler on the R9 device. All stragglers introduce a 40s training delay per local training round (200s per global round).

| Methods | 10% Stragglers | | 20% Stragglers | | 40% Stragglers | |
|---------|----------------|--|----------------|--|----------------|--|
|  | **CIFAR-10** | **FEMNIST** | **CIFAR-10** | **FEMNIST** | **CIFAR-10** | **FEMNIST** |
|  | **MobileNet** | **LEAF** | **MobileNet** | **LEAF** | **MobileNet** | **LEAF** |
| **CFL** | 42 | 90 | 39 | 127 | 39 | 135 |
| **HCFL** | 33 | 64 | 32 | 75 | 34 | 87 |
| **DHS-FL** | 33 | 49 | 32 | 65 | 34 | 67 |
| **DHA-FL** | **5** | **22** | **8** | **30** | **18** | **50** |

*Table 1.* The total convergence time to achieve 60% test accuracy (in minutes) for straggler densities of 10%, 20%, and 40%. In each case DHA-FL handles the stragglers significantly better than the other paradigms, with its strongest comparison at 10% where only a single slow device minimally affects it while the other paradigms already approach worst-case convergence performance .

parity was found between CFL and DHA-FL at 10% stragglers in CIFAR-10 where DHA-FL performed roughly 8x faster. The location of the stragglers within the network had minimal impact on performance across random, even, and concentrated distributions, rather the affected edge server number was found to be the most significant factor.

## 5 CONCLUSION

This paper proposes and demonstrates DHA-FL, a new FL paradigm that can effectively improve the scalability, robustness, and straggler tolerance of the emerging AIoT systems. It achieves this by leveraging the hierarchical networking and computing infrastructure, allowing it to greatly reduce its communication overhead for the same number of working devices. Such a feature is combined with a decentralized asynchronous edge model to improve its resilience to stragglers. Our experiments in the real-life wireless IoT network demonstrate a significant gain in the convergence performance of the proposed solution.

In our future work, we will continue to build upon DHA-FL by incorporating reinforcement learning to enhance adaptability of FL and further explore the operational options of DHA-FL and demonstrate its advantages over commonly-practiced other machine learning optimizations.

## ACKNOWLEDGEMENTS

This work is funded by Intel/NSF joint grant 2003198 and NSF 2008447.

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
