# OpenReview forum: "DHA-FL: Enabling Efficient and Effective AIoT via Decentralized Hierarchical Asynchronous Federated Learning"
_MLSys/2023/Workshop/RCLWN — MLSys-RCLWN 2023_

### Official Review · Reviewer_7xTR · 2023-04-25

**Rating:** 4
**Confidence:** 3

**Review:**

The paper presents a decentralized hierarchical federated learning scheme aimed at decreasing the convergence time of federated learning when devices are heterogeneous.  The proposed DHA-FL scheme consists of two layers: edge servers and workers. In each training cycle, workers update their model and share it with their edge server, which then aggregates the updates from workers and obtains new weights. The edge server then shares the updated weights with its neighbors, and finally, edge servers update their model with model weights coming from adjacent edge servers. The final update step between edge servers is designed to be asynchronous to compensate for the slow or lagging workers within some edge servers. The authors conducted experiments with real hardware and demonstrated a higher convergence rate compared to other methods.

While the problem described in the paper is novel, there are some doubts about the solution method. The experiment only considers the case where 10%, 20%, and 40% of edge servers contain all the slow or lagging workers, which may not be representative of real-world scenarios. Additionally, the proposed method does not solve the problem within the edge server due to slow or lagging workers. If slow or lagging workers are present in all edge servers, then the proposed method may not be effective. One way of solving this problem would be to consider asynchronous model updates within the edge servers and synchronize the model updates between edge servers. Finally, the authors did not compare their proposed method with other asynchronous approaches in Federated Learning, such as [1,2], which consider asynchronous updates from each worker with or without hierarchy.

### Pros:

- Real-world experiments
- Hierarchical system suited for IoT devices

### Cons:

- The proposed solution has some flaws
- Benchmarks used in experiments are not well suited for this problem and should compare against stronger benchmarks
- Experiment only considers a limited scenario where a certain percentage of edge servers contain all the slow or lagging workers, which may not be representative of real-world scenarios

Overall, the paper presents an interesting approach to the problem of decreasing the convergence time of federated learning in the presence of heterogeneous devices. However, further investigation is needed to fully evaluate the proposed method's effectiveness in real-world scenarios.

[1]  Wang, Xing, and Yijun Wang. ‘Asynchronous Hierarchical Federated Learning’. ArXiv [Cs.LG], 2022, http://arxiv.org/abs/2206.00054. arXiv.

[2] Xu, Chenhao, et al. ‘Asynchronous Federated Learning on Heterogeneous Devices: A Survey’. CoRR, vol. abs/2109.04269, 2021, https://arxiv.org/abs/2109.04269.

---

### Official Review · Reviewer_WMvg · 2023-04-30
**Review of a Decentralized Hierarchical Asynchronous Federated Learning (DHA-FL) algorithm**

**Rating:** 7
**Confidence:** 4

**Review:**

This work develops an efficient framework for Decentralized Federated Learning (DFL) in a AI of Things (AIoT) setup. The main objective of this work is to achieve a scalable, robust and fast DFL mechanism. This is achieved through a hierarchical, asynchronous weight-update scheme. Each edge-server at level 2 follows a Centralized FL (CFL) mechanism to update its learnable weights through a fixed set of IoT workers at level 1. The edge-servers then communicate with their 1-hop neighbors to share these updated weights. Finally, the received weights are aggregated locally at each edge-server to complete a learning epoch. This aggregation is asynchronous such that it is performed with as many neighborhood weights as are received until that point without waiting for all the neighbors to communicate.  A final global consensus update is performed at a global aggregation server after a fixed number of epochs.

Following are the key features of the proposed algorithm:
1) It is scalable on account of the hierarchical scheme since adding more users at level 1 does not affect the inter-server communication overhead at level 2 which is fixed by the number of edge-servers.
2) It is naturally more robust than CFL due to less dependance on a particular centralized-server. In DFL, even if one edge-server is affected, the rest can can continue to learn.
3) It is also fast since the asynchronous update scheme reduces dependance on slow workers or servers.

Strengths:
1) Main impact of this work is experimentally illustrated by faster convergence for the cases wherein at least one server is lagging behind others (straggler). Clearly this is due to the asynchronous update mechanism. Another key experimental observation is that it is the number of stragglers in the network and not their specific locations that impact the overall convergence time of the proposed algorithm.
2) Experimental analyses are strong, validate the authors' claims, and are presented on real wireless network scenario which can be a nice benchmark for future works. This is the main motivating factor for me to recommend an accept.
3) Write-up is cogent and easy to follow. The algorithm is described with adequate clarity and the experimental setup is sufficiently detailed for reproducibility.

Room for improvement:
1) Proposed method is not fully decentralized as it consists of a centralized setup for learning the edge-server weights from its workers. Clearly, one edge-server being affected can take its entire set of workers out of the learning mechanism. Therefore, the overall robustness of the algorithm is questionable. The global aggregation is again a centralized mechanism which entails the same issues.
2) Novelty of this work is limited. All the key aspects viz. decentralization, hierarchical communication and asynchronous updates are well studied in context of FL. Although the proposed assembly for the chosen application area is moderately novel in my opinion.
3) The authors claim, that for results in Table 2, the convergence times for all the baselines approach the worst-case at 10% straggler density and therefore remain fixed at higher densities. Whereas, the proposed DHA-FL becomes progressively worse with increase in stragglers. This is not fully clear to me. For CIFAR, it seems that this could be a legitimate analysis, although if we take CFL to be the worst-case performance then even that argument doesn't hold. Moreover, for FEMNIST the convergence time becomes progressively worse for all methods across higher straggler densities and the change in convergence time between 20% and 40% for DHS-FL is smaller than that of DHA-FL. And clearly, both are pretty far from worst-case at that point. Therefore, I feel that these results require a better discussion than what is provided to definitively establish the utility of DHA-FL.
4) Their are some minor typos in the write-up. For example, at column 2 line 81, the model queue has been denoted by 'w' which should be 'W' instead. Also, there is a typo in column 1 line 204. Please revise the manuscript to correct such typos.

---

### Official Review · Reviewer_cDVP · 2023-04-30
**DHA-FL**

**Rating:** 7
**Confidence:** 3

**Review:**

The paper presents a decentralized hierarchical approach to improve scalability, robustness and resilience in federated learning. In particular, the approach presented in the paper relies on a two-stage model aggregation algorithm: in the lower abstraction level, each edge server updates its edge model by aggregating information from its assigned workers using a centralized federated learning approach, and then communicates with neighboring edge servers to update the shared global model. To mitigate the effects of stragglers, each edge server uses an asynchronous model averaging approach to update its model based on information received from its neighbors without waiting for slow servers. To illustrate the benefits of the proposed framework, the paper also tests the proposed approach in a live multi-hop network.

Overall the paper is well-written, easy to follow, and addresses a topic of interest to the conference audience. The approach presented in the paper is, to the best of my knowledge, novel, and the numerical experiments on the live testbed show that the proposed approach does lead to improved performance when compared against classic federated learning approaches. Hence I believe the paper should be accepted for presentation. Some questions and comments follow.

1.	In the third paragraph on the first page, the authors mention that decentralized federated learning provides a more robust learning framework. I believe it would be beneficial to expand upon that claim, explaining why DFL improves robustness, and corroborating it with some more specific references.

2.	The paper claims that the proposed approach addresses robustness, scalability and straggler challenges. I can see more clearly how the proposed approach addresses scalability and stragglers, but is the claimed improved robustness a byproduct or using a decentralized approach? Or does the hierarchical nature of the algorithm proposed here does improve robustness?

3.	Would it be possible to add some experiments with different degrees of heterogeneity? It would be interesting to see how that impacts the convergence of the proposed approach.

---

### Meta-Review · Area_Chair_UxG5 · 2023-05-08

**Recommendation:** Accept
**Confidence:** 4

**Metareview:**

Reviewers identified this paper as well-written with clear results and benefits in the inclusion of asynchroncity in the DHFL framework to improve robustness against straggling workers. The results in a live testbed were also highlighted as a strong aspect of the paper.

The reviewers did point out some areas of improvement regarding clarifications on the nature of the robustness of the proposed approach, clarity on novelty/comparisons against existing asynchronous methods, and testing the method on more interesting cases in which lagging workers are distributed across the servers. Some additional clarity on the numerical results is also recommended.